# Body Composition, Somatotype and Raw Bioelectrical Impedance Parameters of Adolescent Elite Tennis Players: Age and Sex Differences

**DOI:** 10.3390/ijerph192417045

**Published:** 2022-12-19

**Authors:** Antonio J. Berral-Aguilar, Susana Schröder-Vilar, Daniel Rojano-Ortega, Francisco J. Berral-de la Rosa

**Affiliations:** 1CTS-595 Research Group, Department of Informatics and Sports, University Pablo de Olavide, 41014 Seville, Spain; 2Tennis Empowerment Center (T.E.C.) Carles Ferrer Salat, Can Marlés, 08960 Barcelona, Spain

**Keywords:** body composition, somatotype, tennis, bioimpedance, phase angle, resistance, reactance

## Abstract

Studies analyzing tennis players’ body composition and morphological and bioelectrical characteristics are scarce, especially among adolescents. This study aimed to explore sex- and age-based differences in body composition, somatotype, and bioelectrical properties among elite young male and female tennis players aged 13–16 years. Twenty-two male (14.45 ± 1.10 years) and 20 female (14.30 ± 1.03 years) elite tennis players participated in this study and were compared according to sex (males vs. females) and, within each sex, according to age (13–14 years vs. 15–16 years). Female adolescent elite tennis players had higher body fat (BF) percentage and higher endomorphy than males. They also had lower skeletal muscle mass and total body water (TBW) percentages. Older boys had lower resistance and a higher TBW and phase angle (PhA) than younger boys, likely due to maturation and performance. No significant differences were found between younger and older girls, except for the PhA, potentially associated with better cell function and performance. This study provides valuable reference data for coaches of elite youth tennis players. Due to the few differences found in body composition and somatotype in the different age groups, the PhA may be used by practitioners as a reference for cell function and performance.

## 1. Introduction

Tennis is a popular racket sport with over 87 million players worldwide [1]. The Spanish Ministry of Culture and Sports [2] indicates that, according to federation license data, tennis is Spain’s ninth most popular sport. It requires high performances in speed, agility, strength, power, and many other abilities [3]. In recent decades, tennis has evolved from a primarily technical sport to a more explosive and physically demanding one [4].

Body composition is considered a determinant of athletic performance. It describes the amount of the various components of the human body [5]. Body composition can be determined at different levels. Two of them are the cellular level, which includes adipose cells, body cell mass, and intracellular and extracellular water, and the tissue level, which includes the adipose, lean soft, and skeletal muscle mass tissues [5,6]. Parameters from different levels should not be combined for the accurate assessment of body composition, but a separate evaluation may provide practical information [7].

Many studies have shown that competing at the highest level is closely associated with body composition, anthropometric characteristics, and somatotype [8,9,10]. Significant differences in body composition have been observed among athletes from different sports and even from different playing positions in the same sport [11]. Furthermore, body composition and somatotype have evolved parallel with sport’s evolution [12].

Somatotype represents the athlete’s morphological conformation and is usually calculated by the Heath and Carter method [13]. It consists of three components: endomorphy, which represents relative adiposity; mesomorphy, which represents musculoskeletal robustness; and ectomorphy, which represents the relative linearity of the human body [13].

The body composition and somatotype of the best elite athletes are considered patterns to which particular athletes’ physical characteristics should be as close as possible [14]. Therefore, interest in anthropometric characteristics, body composition, and somatotypes has increased over the past two decades. Specific profiles have even been proposed as useful measures for talent detection in many sports [15,16,17]. However, most studies have focused on sports other than tennis.

Bioelectrical impedance analysis (BIA) is a safe, non-invasive method for estimating body composition. BIA devices introduce a weak, alternating current into the human body, detect the voltage drop, and calculate electrical parameters [18]. These parameters are then introduced into specific equations to determine body composition [19]. However, these equations are only accurate to specific populations and under certain circumstances [20].

Resistance and reactance are two electrical parameters provided by BIA which are uninfluenced by the equations that may affect body composition compartments [21,22]. A bioelectrical phase angle (PhA) is calculated from them as the reactance to resistance ratio’s arctangent. The PhA is considered an indicator of cellular health, cell membrane integrity, and cell function [23,24] and is influenced by physical activity level [25]. In sports, higher PhA values have been associated with better performances [26,27]. In addition, Martins et al. [28] reported that a PhA could be used to monitor adolescents’ physical fitness. Many studies have validated BIA’s accuracy for estimating body water compartments [29], and according to Francisco et al. [30], directly measured raw BIA variables (resistance, reactance, and the PhA) are useful predictors of cellular hydration and fluid distribution in athletes.

There are several BIA technologies but, to date, the foot-to-hand technology at 50 kHz single frequency is acknowledged as the reference method in humans [31,32], and Campa et al. [7] established some general recommendations for BIA application in athletes. Although BIA-derived body composition parameters are strongly influenced by the predictive equations used, a recent systematic review reported that BIA can be considered a valid method for assessing body composition in athletes provided that foot-to-hand technology and specific equations for athletes are used [33]. However, those equations were not specific for adolescents, and, therefore, we only used BIA in this study to determine resistance, reactance, the PhA, and total body water (TBW), but the rest of the body composition parameters were calculated from anthropometric measures.

Adolescence is a human developmental state characterized by multiple changes in body composition [34]. While body composition and somatotype have frequently been associated with performance, few studies have investigated changes in adolescent elite tennis players’ body composition and somatotype. To our knowledge, only two recent studies have analyzed morphology and body composition in young tennis players. However, apart from height and body mass, one of them determined only body fat (BF) and slim muscle mass [35], and the other only body mass index (BMI) and BF [36]. National and international competitions for young tennis players are generally structured into chronological age-based groups. Therefore, this study had three aims: (1) to assess sex differences in body composition, somatotype, and raw bioelectrical impedance parameters among elite young male and female tennis players aged 13–16 years; (2) to evaluate the effects of chronological age on those variables; and (3) to provide reference data for elite tennis players of the same age. We hypothesized that most measured variables would differ between boys and girls and between age groups.

## 2. Materials and Methods

### 2.1. Participants

Forty-two adolescent Spanish tennis players aged 13–16 years old volunteered to participate in this study (22 males and 20 females). All had a minimum of 4 years of experience, and, to be considered elite tennis players, they had to be classified in the top 30 of their respective categories when the study began. None of the participants had experienced serious injuries at least three months before the testing sessions. All the participants’ parents or legal tutors provided written informed consent according to the Declaration of Helsinki. This study was approved by the Research Ethics Committee of two Spanish hospitals (C.P. PIS-CC C.I. 2061-N-21). The participants were compared according to sex (males vs. females) and, within each sex, according to age (13–14 years vs. 15–16 years). The mean age, height, and weight of each group are provided in Table 1.

### 2.2. Procedures

Anthropometric measurements were performed following the Spanish Group of Kinanthropometry guidelines [37] and according to the International Society for Advancement in Kinantropometry (ISAK) [38], by the same anthropometrist accredited by the Andalusian Medical Association for Physical Education and Sport, which follows ISAK guidelines. The study was performed in the Tennis Empowerment Center (T.E.C.) Carles Ferrer Salat, three months after the beginning of the season. All measurements were taken on the participant’s right side, between 10:00 and 12:00, the day before the first T.E.C. Cup’s competition. Body height was measured to the nearest 0.1 cm with a stadiometer (SECA, Hamburg, Germany), and body weight was measured to the nearest 0.1 kg with the body composition analyzer (Tanita MC-780MA, Tokyo, Japan). Skinfolds were taken to the nearest 0.5 mm using a caliper (SATA, Seville, Spain), girths were measured to the nearest 0.1 cm with a flexible metallic tape measure (SATA, Seville, Spain), and breadths were measured to the nearest 0.1 mm with a short branch pachymeter (SATA, Seville, Spain). All measurements were taken three times, and the average value was used in subsequent calculations. The technical error of measurement ranged from 0.22% to 1.12%.

Anthropometric variables included body mass, body height, four skinfolds (triceps, subscapular, supraspinal, and medial calf), four girths (upper arm relaxed and flexed/tensed, mid-thigh, and maximal calf), and two breadths (femoral and humeral). BMI was calculated as weight/height^2^, where weight was expressed in kg and height in m. BF percentage was estimated using the equations of Slaughter et al. [39], and skeletal muscle mass percentage (SMM) was determined using the method of Poortmans et al. [40]. Somatotype was determined according to the equations of Carter and Heath [13] and plotted on a two-dimensional somatochart with Microsoft Office Excel 2016 (Seattle, WA, USA).

BIA was performed with a segmental multi-frequency body composition analyzer (Tanita MC-780MA, Tokyo, Japan) by the same professional. The evaluation was undertaken between 10:00 and 12:00, after an overnight fast of at least 8 h, the day before the T.E.C. Cup’s competition. The participants had refrained from intense physical exercise 12 h before testing and were evaluated with an empty bladder. Measurements were taken in a standing position using 8 electrodes according to the manufacturer’s guidelines: 4 were in contact with the soles and heels of both feet, and 4 were in contact with the palms and fingers of both hands. The whole body resistance, reactance, and PhA at 50 kHz, as well as TBW, were calculated. The coefficient of variation determined previously with thirteen participants in our laboratory for the resistance, reactance, and PhA at 50 kHz, as well as for total body water, ranged from 1.15 to 2.92%.

### 2.3. Statistics

Statistical analyses were performed using SPSS for Windows, v. 22.0 (SPSS Inc., Chicago, IL, USA). The means and standard deviations of all variables were calculated. Data were tested for normality using the Shapiro–Wilk test. When this condition was met, the Student’s *t*-test was used to assess significant differences between groups. When the normality condition was not met, the Mann–Whitney U test was performed. Results were considered statistically significant if *p* < 0.05. The magnitude of the differences between groups was assessed using Hedge’s *g* with the correction for small sample sizes, and they were interpreted according to the following criteria: minimal effect (<0.20), small effect (0.20–0.50), moderate effect (0.50–0.80) or large effect (>0.80) [41]. Post-hoc power calculations were performed with GPower v. 3.1 (Düsseldorf, Germany).

## 3. Results

Descriptive statistics for all variables obtained in the boys and girls groups are presented in Table 1. Significant differences between them, effect sizes, and observed statistical power are also provided in Table 1.

Boys showed significantly lower BF and endomorphy values but significantly higher SMM and TBW values than girls (*p* < 0.01), all with a large effect size.

Descriptive statistics for all variables obtained in the younger (13–14 years) and older (15–16 years) subgroups of boys and girls are shown in Table 2 and Table 3. Significant differences between age groups, effect sizes, and observed statistical power are also provided in Table 2 and Table 3.

In addition to the expected differences in age, body weight, height, and BMI, the boys’ younger subgroup had significantly lower TBW (*p* < 0.05) and PhA (*p* < 0.01) values and a significantly higher resistance (*p* < 0.01) value than the older subgroup, with moderate to large effect sizes. With regard to the girls, in addition to the expected differences in age, body weight, and height, the girls’ younger group had a significantly lower PhA value (*p* < 0.05) than the older subgroup, with a large effect size.

Figure 1 illustrates the somatotypes for the different groups. The mean somatotype could be defined as mesomorphic-ectomorphic (1.69–3.72–4.11) for all boys and mesoectomorphic (2.45–3.16–3.80) for all girls. The mean somatotype could be defined as mesoectomorphic (1.96–3.53–4.46) for the younger subgroup of boys and mesomorphic-ectomorphic (1.37–3.93–3.69) for the older subgroup. Concerning the girls, the mean somatotype was mesoectomorphic for the younger (2.47–3.18–3.94) and older (2.42–3.13–3.67) subgroups.

## 4. Discussion

The first purpose of this study was to identify sex differences in body composition, somatotype, and bioelectrical impedance parameters in a group of elite male and female tennis players aged 13–16 years. As we hypothesized, some of the variables measured differed between groups. Girls had significantly higher BF values and significantly lower SMM values than boys. Moreover, because endomorphy represents relative fatness [13,14], they also had higher endomorphy values.

Luna-Villouta et al. [35] also observed differences in BF and SMM percentages between a group of 58 male tennis players aged 15.4 ± 0.8 years and a group of 29 female tennis players aged 15.3 ± 0.8 years. These sex differences in body composition and somatotype were expected and have also been found between boys and girls of similar ages in other sports or in non-athlete populations [42,43,44].

In addition to these differences, girls also had significantly lower TBW values than boys, which is consistent with previous studies, even on non-athletes [29,45,46,47]. These differences may be explained by boys’ lower adiposity levels [46,47] since girls had higher BF (18.94 ± 3.42%) than boys (14.47 ± 3.54%) in this study.

Not all studies investigating body composition and somatotype among tennis players of approximately the same age obtained similar values for the measured variables. Sánchez-Muñoz et al. [48] reported much higher BF and endomorphy values in a group of elite junior tennis players aged 14–16 years. The lower mean age of our players could explain these differences. However, when we stratified our groups by age, we also observed them in our older (15–16 years) subgroups. Therefore, the only possible explanation for these differences is a change in coaching style. Historically, tennis coaches did not closely monitor players’ nutritional aspects. However, modern coaches focus not only on the technical aspects but also on nutritional strategies to reduce BF. This view is supported by recent studies on youth tennis players, which obtained BF and SMM values highly similar to ours [35,36]. Indeed, this evolution to a lower BF and endomorphy has also been observed in other sports, such as soccer [12,49].

The second purpose of this study was to assess age differences between age-based subgroups (13–14 years vs. 15–16 years). Among boys, the younger subgroup had significantly lower TBW value than the older subgroup, potentially due to their non-significantly higher BF value [46,47]. This result is similar to that observed by Toselli et al. [50], who analyzed a group of pre-adolescent soccer players and found that those with advanced maturity status had higher TBW.

No other body composition or somatotype parameters differed significantly between the boys’ subgroups, which is consistent with the research of Luna-Villouta et al. [35], who showed a non-significant correlation between chronological age and BF and a significant but weak correlation between chronological age and SMM in a group of young male tennis players. Similarly, other studies performed with youth soccer or basketball players found no significant differences in BF or somatotype between different age groups [51,52,53].

However, concerning bioelectrical parameters, resistance was significantly higher in the younger subgroup, which was somewhat expected since they had also lower TBW values. Resistance is a measure of different tissues’ resistivity when conducting electrical current and is calculated from the voltage decrease as the current passes through body fluids, which are ionic solutions [23,54]. Therefore, increased body resistance is associated with decreased TBW [30,55].

The PhA was significantly lower in the younger subgroup, which is consistent with the results found by Koury et al. [56], who compared adolescents of different sports. In previous studies, the PhA positively correlated with TBW [57] and physical maturation [58], which may explain the significantly higher value observed in our older subgroup. In addition, higher PhA values have been associated with better performances [26,27], likely found in the older boys.

Among girls, only the PhA differed significantly between subgroups, and this difference alone was unexpected. The absence of differences in the other measured variables may indicate that girls aged ≥13 years had already reached their physical maturation. Indeed, Sögüt et al. [36] calculated girls’ physical maturation as a percentage of their predicted adult stature, which most girls aged ≥13 years had already reached. These results are in agreement with those found by Luna-Villouta et al. [35] in a group of young female tennis players.

Ferreira et al. [58] concluded that mature athletes have higher values than non-matured athletes. Therefore, since our girls had likely already reached their physical maturation, significant PhA differences may be associated with better cell function and performance [24,26,27]. It would be interesting to study this relationship in the future.

As expected, the body composition and somatotype values observed in our group of tennis players differ from those observed in other sports. For example, BF and endomorphy are lower in male tennis players than in basketball players of the same age, while their mesomorphy and ectomorphy are similar [53,59]. BF is usually lower and mesomorphy is usually higher in male and female soccer players [8,52,60]. Furthermore, since no other studies have determined raw bioelectrical impedance parameters in adolescent tennis players, our study may serve as a reference for coaches of elite tennis players of similar ages, which was the third purpose of this study.

This study has three relevant limitations. Firstly, sample sizes for the different groups were small, compromising the statistical power of some results. Secondly, while its results can be used as a standard reference, they should be interpreted with caution, because participants had very similar levels. Therefore, their transfer to other levels should be made according to individual characteristics and necessities. Lastly, recent studies have observed a lack of agreement in BIA measurements between supine and standing positions, even in raw bioelectrical parameters [61,62]. Therefore, our reference values should not be compared with those obtained with supine bioimpedance analyzers.

Exploring the resistance, reactance, and PhA in adolescent tennis players is this study’s greatest strength. These variables have not been previously explored in these athletes and may be important for monitoring their body composition and performance. However, future studies should focus on analyzing body composition, somatotype, and bioelectrical parameters, not only as a function of chronological age but also biological maturity. The differences associated with maturity are transitory but very important during adolescence, especially in males aged 14.5 years, in whom early, on-time, and late maturity is observed.

## 5. Conclusions

Based on this study’s results, we conclude that, as expected, female adolescent elite tennis players have higher BF and endomorphy than males. They also have lower SMM, and, due to their greater adiposity, their TBW is also lower. Older boys have lower resistance and a higher TBW and PhA than younger boys, which may be associated with maturation and performance. For the reason that most girls aged ≥13 years have already reached their physical maturation, no significant differences were found between younger and older girls, except for the PhA, which may be associated with better cell function and performance. Coaches may use this study’s data as a reference for advisable values for elite youth tennis players. Body composition is one of the factors influencing sports performance. Due to the few differences found in body composition and somatotype in the different age groups of elite adolescent tennis players, a PhA may be used by practitioners as a reference for cell function and performance. Nevertheless, future studies exploring the relationships between a PhA and performance in tennis players are warranted.

## Figures and Tables

**Figure 1 ijerph-19-17045-f001:**
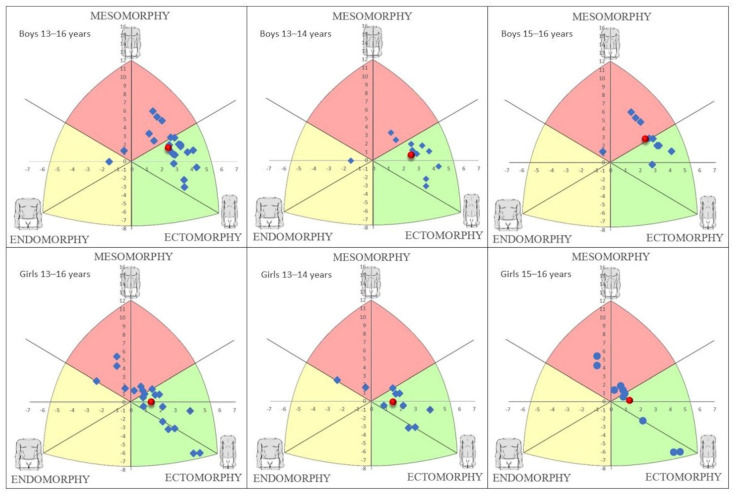
Somatotype distribution in the different groups of boys and girls.

**Table 1 ijerph-19-17045-t001:** Descriptive statistic and differences between the group of boys and the group of girls.

Variables	Boys (n = 22) Mean ± SD	Girls (n = 20) Mean ± SD	*p*-Value	Hedge’s g	Power (1−β)
Age (years)	14.45 ± 1.10	14.30 ± 1.03	0.676	0.14	0.07
Body mass (kg)	48.87 ± 9.87	46.70 ± 8.61	0.454	0.23	0.11
Height (cm)	162.43 ± 8.31	158.70 ± 9.83	0.066	0.40	0.23
BMI (kg m^−2^)	18.33 ± 2.24	18.42 ± 2.18	0.940	−0.04	0.05
BF (kg)	7.13 ± 2.59	8.96 ± 2.93	0.012	−0.64	0.50
BF (%)	14.47 ± 3.54	18.94 ± 3.42	<0.001	−1.25	0.97
SMM (kg)	18.79 ± 3.02	16.27 ± 3.51	0.017	0.75	0.66
SMM (%)	38.81 ± 2.91	34.71 ± 2.17	<0.001	1.55	1.00
Endomorphy	1.69 ± 0.77	2.45 ± 0.69	<0.001	−1.01	0.88
Mesomorphy	3.72 ± 0.86	3.16 ± 1.13	0.074	0.54	0.38
Ectomorphy	4.11 ± 0.99	3.80 ± 1.25	0.378	0.27	0.14
TBW (kg)	29.84 ± 5.84	26.77 ± 4.47	0.065	0.57	0.44
TBW (%)	61.13 ± 2.20	57.53 ± 2.14	<0.001	1.61	1.00
Resistance (Ω)	655.72 ± 89.18	703.32 ± 86.48	0.118	−0.53	0.37
Reactance (Ω)	71.16 ± 7.14	72.96 ± 9.14	0.479	−0.21	0.10
PhA (°)	6.24 ± 0.51	5.95 ± 0.52	0.067	0.55	0.40

Legend: SD: Standard Deviation; BMI: body mass index; BF: body fat; SMM: skeletal muscle mass; TBW: total body water; PhA: phase angle.

**Table 2 ijerph-19-17045-t002:** Descriptive statistic and differences between the two group of boys.

Variables	Boys 13–14 years (n = 12) Mean ± SD	Boys 15–16 years (n = 10) Mean ± SD	*p*-Value	Hedge’s g	Power (1−β)
Age (years)	13.58 ± 0.51	15.50 ± 0.53	<0.001	−3.55	1.00
Body mass (kg)	42.38 ± 6.97	56.67 ± 6.63	0.001	−2.02	0.98
Height (cm)	156.90 ± 7.44	169.06 ± 2.01	<0.001	−2.15	1.00
BMI (kg m^−2^)	17.10 ± 1.49	19.80 ± 2.15	0.001	−1.40	0.86
BF (kg)	6.61 ± 3.08	7.74 ± 1.82	0.043	−0.43	0.15
BF (%)	15.21 ± 4.21	13.58 ± 2.45	0.346	0.46	0.17
SMM (kg)	16.92 ± 2.58	21.02 ± 1.75	0.000	−1.79	0.98
SMM (%)	40.00 ± 1.16	37.38 ± 3.75	0.052	0.91	0.53
Endomorphy	1.96 ± 0.92	1.37 ± 0.35	0.093	0.82	0.43
Mesomorphy	3.53 ± 0.80	3.95 ± 0.92	0.267	−0.47	0.18
Ectomorphy	4.46 ± 0.80	3.69 ± 1.08	0.069	0.78	0.41
TBW (kg)	25.61 ± 3.75	34.92 ± 3.12	<0.001	−2.60	1.00
TBW (%)	60.58 ± 2.35	61.78 ± 1.92	0.043	−0.54	0.22
Resistance (Ω)	710.98 ± 69.91	589.40 ± 62.38	<0.001	1.77	0.97
Reactance (Ω)	73.91 ± 7.55	67.86 ± 5.21	0.054	0.90	0.52
PhA (°)	5.94 ± 0.18	6.61 ± 0.54	0.003	−1.60	0.94

Legend: SD: Standard Deviation; BMI: body mass index; BF: body fat; SMM: skeletal muscle mass; TBW: total body water; PhA: phase angle.

**Table 3 ijerph-19-17045-t003:** Descriptive statistic and differences between the two group of girls.

Variables	Girls 13–14 years (n = 10) Mean ± SD	Girls 15–16 years (n = 10) Mean ± SD	*p*-Value	Hedge’s g	Power (1−β)
Age (years)	13.40 ± 0.52	15.20 ± 0.42	<0.001	−3.65	1.00
Body mass (kg)	41.74 ± 8.66	51.66 ± 5.19	0.006	−1.33	0.80
Height (cm)	153.30 ± 7.29	164.09 ± 9.27	0.002	−1.24	0.72
BMI (kg m^−2^)	17.61 ± 2.24	19.23 ± 1.90	0.075	−0.75	0.34
BF (kg)	8.27 ± 4.04	9.66 ± 0.85	0.023	−0.46	0.16
BF (%)	19.13 ± 4.74	18.75 ± 1.44	0.811	0.10	0.06
SMM (kg)	14.52 ± 3.63	18.02 ± 2.45	0.015	−1.08	0.60
SMM (%)	34.59 ± 1.79	34.84 ± 2.59	0.806	−0.11	0.06
Endomorphy	2.47± 0.83	2.42 ± 0.57	0.879	0.07	0.05
Mesomorphy	3.18 ± 0.76	3.13 ± 1.46	0.684	0.04	0.05
Ectomorphy	3.94 ± 1.08	3.67 ± 1.45	0.646	0.20	0.07
TBW (kg)	24.04 ± 4.33	29.50 ± 2.64	0.003	−1.46	0.87
TBW (%)	57.85 ± 1.96	57.20 ± 2.36	0.511	0.29	0.09
Resistance (Ω)	724.56 ± 79.18	682.07 ± 92.28	0.280	0.47	0.16
Reactance (Ω)	71.23 ± 8.53	74.69 ± 9.85	0.412	−0.36	0.12
PhA (°)	5.62 ± 0.33	6.27 ± 0.48	0.002	−1.51	0.89

Legend: SD: Standard Deviation; BMI: body mass index; BF: body fat; SMM: skeletal muscle mass; TBW: total body water; PhA: phase angle.

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
