# Peer review of "Body Composition, Somatotype and Raw Bioelectrical Impedance Parameters of Adolescent Elite Tennis Players: Age and Sex Differences"

_ijerph, 2022, doi:10.3390/ijerph192417045_

Round 1

Reviewer 1 Report

General comments

- The assessment of body mass components is an important challenge nowadays. However, the Berral-Aguila and coworkers do not properly consider the most important theoretical concepts underlying the basics of body composition analysis. Additionally, body composition differences due to maturity status were not addressed. Especially in male adolescents of 14.5 y where early, on time, and late maturity subjects can be found. Considering the selected sample size, the aforementioned points represent the main limitations of the present study. 

- While I understand that the authors are not native English, a revision of the manuscript is fundamental to improve its readability. 

Specific comments

Abstract:

- “This study aimed analyzing tennis players’ body composition and morphological and bioelectrical characteristics are scarce, especially in adolescents.” This sentence does not make sense. 

Introduction

- Lines 48-49: “Resistance and reactance are two electrical parameters provided by BIA, that are uninfluenced by the equations and may affect body composition compartments” Resistance and reactance (R and Xc, respectively) cannot affect body composition compartments. Maybe they can reflect body mass components. Please, revise. 

- The introduction needs to be clear what the practical question is that you are trying to address. How the answer to this question is important to the field as this is not clear or obvious? How is this study and impactful study and not trivial as this needs more clarity as well. The key issue here is to make sure you set up your approach to the problem. The approach to the problem is essential in determining and describing the rationale for the study. You have not given a basic rationale for the choices made for the variables (somatotype, raw BI parameters, and BIA-derived body components) used in the study. 

- Several key studies from the past 3 years regarding bioelectrical impedance analysis in athletes are missing. Check the MPDI literature too. 

Methods

- Authors should provide a more detailed description of the operation and validity of the impedance measurements. Measuring bioelectrical impedance with direct segmental technique in a standing position can lead to a gravitational effect on body fluids making some artefact in the estimated body composition measures. In fact, a recent systematic review (EJAP 2021) higlight the lack of predictive equations for athletes for the BIA in standing position. Then, there is a basic need to describe the technical characteristics of the TANITA device. Describe the probe and sound frequency (does it work in mono or multifrequency?). What is the calibration method to ensure validity (accuracy and precision) of the BIA measurements? What is the technical error of measurement in vivo? Provide readers with a concise description of what this device measures.

- Also, what are the measurements detected by this tool? Do they directly measure the raw bioimpedance parameters (e.g. R, Xc and PhA)? Again, what equation was used to estimate the body composition variables? are they equations developed using the same TANITA device or an instrument that works with similar characteristics (frequency and measurement technique)? 

- The use of predictive equations developed for the general population may militate against the quality of the presented data. It is necessary to add more info about the formulas used in this study.

Results

- Tables. Please, add absolute values of FM, TBW, and SMM as well. 

BW (cm should be kg). Also, is It really necessaire to report such as many acronymous? BW could be reported as body mass. Inexpert readers would benefice from that. 

SMM: what does “s” mean in (%s)?

Discussion

- Considering the abovementioned limitations, it seems that body composition differences among tennis players stratified according to gender and chronological age are the main finding of this study. Similarly, how do practitioner benefit from that? Again, the discussion section fails to relate the findings to this particular application of interest. 

- The discussion section is very descriptive and offers limited comparisons to previous research. it is important to consider that the BI and BI-estimated body composition parameter is a dependent instrument and that the instrumental sensitivities are different. Therefore, no comparisons can be made between studies that measure bioelectrical proprieties with different devices. Especially, if predictive equations are not listed in the methods section and their performance addressed and discussed over the text. 

- Authors are therefore encouraged to make substantial changes throughout to improve the overall quality. In the current form, the rationale for the study is not clear, the new value is unclear, and I have difficulties finding specific take-home messages for practitioners.

Reviewer 2 Report

1. Despite considering the sample size to be short, I understand that another statistical technique could be used. There must be calculation of the sample power with GPower or, alternatively, use non-parametric statistics.

2. The phase of the season in which the data collection was carried out is not well identified;

3. I understand that the conclusions say little about purpose of the studyband there is no reference to future implications.

Tank you 

Author Response

Changes in the manuscript are made in green

1. Despite considering the sample size to be short, I understand that another statistical technique could be used. There must be calculation of the sample power with GPower or, alternatively, use non-parametric statistics.

We are aware that an a priori power test should have been performed. As many other authors, we are not sure of the importance of calculating post-hoc power, but, since this reviewer requested it, we have included it in the tables so that readers can see the observed power of the effect sizes found.

2. The phase of the season in which the data collection was carried out is not well identified;

The phase of the season has been included.

3. I understand that the conclusions say little about purpose of the study and there is no reference to future implications.

The conclusions have been improved and future implications have been now indicated in the conclusions part and some lines before.

Reviewer 3 Report

The subject matter is relevant to readers of this journal, the rationale is consistent, and the approach is interesting. Body composition is acknowledged as a determinant of athletic health and performance. The phase angle is a parameter that can be used to monitor the progression of a disease or the effectiveness of an intervention. Values above or below reference values may be helpful in patient care and clinical outcomes. Therefore, I think this manuscript represents a worthwhile and significant contribution to the body of literature.

INTRODUCTION

[Concern 1]

At the end of the introduction section the objective is clear. Based on the literature review, authors did not clarify the hypothesis of the study. Please include.

METHODS

[Concern 2]

Due to the small sample size, sample power calculation is required.

[Question 1]

Study Protocol: when were all assessments perform? What time of the season? Please explain.

RESULTS

[Concern 3]

In table 2, the p values = 0.000 should be presented as <0.001.

The conclusion is weak and focused on future directions. Please rewrite this section.

Author Response

Specific changes in the manuscript are made in blue or green (green if the second reviewer also requested them)

INTRODUCTION

[Concern 1]

At the end of the introduction section the objective is clear. Based on the literature review, authors did not clarify the hypothesis of the study. Please include.

The hypothesis has been included.

METHODS

[Concern 2]

Due to the small sample size, sample power calculation is required.

We are aware that an a priori power test should have been performed. As many other authors, we are not sure of the importance of calculating post-hoc power, but, since this reviewer requested it, we have included it in the tables so that readers can see the observed power of the effect sizes found.

[Question 1]

Study Protocol: when were all assessments perform? What time of the season? Please explain.

This information has been included.

 RESULTS

[Concern 3]

In table 2, the p values = 0.000 should be presented as <0.001.

It has been changed.

The conclusion is weak and focused on future directions. Please rewrite this section.

The conclusion has been improved and future directions have been now indicated in the conclusions part and some lines before.

Round 2

Reviewer 1 Report

I have additional requests in order to improve the overall quality. 

BIA guidelines for athletes were not mentioned: Assessment of Body Composition in Athletes: A Narrative Review of Available Methods with Special Reference to Quantitative and Qualitative Bioimpedance AnalysisNutrients 2021, 13(5), 1620; https://doi.org/10.3390/nu13051620. Also, the authors are encouraged to read this paper to better explain body mass compartmental models, which represent the basics of body composition that should be presented in the introduction section. 

In the first round of revisions, It was suggested to consider a recent systematic review where BIA-based body mass estimations were compared to BC data obtained with reference methods. It is very important to highlight that BIA can be considered a valid method for predicting body mass components, provided that specific procedures for athletes are considered.  

Lastly, it should be reinforced the concept that different BIA technologies provided different outputs resulting in a lack of agreement between foot-to-hand and direct segmental (BIA in standing position) or BIS.

Author Response

Changes in the manuscript are made in red

BIA guidelines for athletes were not mentioned: Assessment of Body Composition in Athletes: A Narrative Review of Available Methods with Special Reference to Quantitative and Qualitative Bioimpedance AnalysisNutrients 2021, 13(5), 1620; https://doi.org/10.3390/nu13051620. Also, the authors are encouraged to read this paper to better explain body mass compartmental models, which represent the basics of body composition that should be presented in the introduction section. 

A new paragraph with three new references has been included in the introduction. One of the new references is the narrative review mentioned above.

In the first round of revisions, It was suggested to consider a recent systematic review where BIA-based body mass estimations were compared to BC data obtained with reference methods. It is very important to highlight that BIA can be considered a valid method for predicting body mass components, provided that specific procedures for athletes are considered. 

Another new paragraph with new references has been included in the introduction. One of these references is the recent systematic review mentioned above.

Lastly, it should be reinforced the concept that different BIA technologies provided different outputs resulting in a lack of agreement between foot-to-hand and direct segmental (BIA in standing position) or BIS.

This has been included in the limitations.